# Association between Dietary Patterns Reflecting C-Reactive Protein and Metabolic Syndrome in the Chinese Population

**DOI:** 10.3390/nu14132566

**Published:** 2022-06-21

**Authors:** Huan Yu, Qiaorui Wen, Jun Lv, Dianjianyi Sun, Yuan Ma, Sailimai Man, Jianchun Yin, Mingkun Tong, Bo Wang, Canqing Yu, Liming Li

**Affiliations:** 1Department of Epidemiology and Biostatistics, School of Public Health, Peking University Health Science Center, Beijing 100191, China; yuhh@pku.edu.cn (H.Y.); wqr119@163.com (Q.W.); epi.lvjun@vip.163.com (J.L.); dsun1@bjmu.edu.cn (D.S.); sailimai.man@meinianresearch.com (S.M.); lmleeph@vip.163.com (L.L.); 2Peking University Center for Public Health and Epidemic Preparedness and Response, Peking University, Beijing 100191, China; 3Peking University Health Science Center Meinian Public Health Institute, Beijing 100191, China; yuan.ma@meinianresearch.com (Y.M.); tongmk@bjmu.edu.cn (M.T.); 4MJ Health Care Group, Shanghai 200041, China; yinjc@health-100.cn

**Keywords:** metabolic syndrome, dietary patterns, C-reactive protein, cross-sectional study

## Abstract

It is unclear how the dietary patterns reflecting C-reactive protein (CRP) affect metabolic syndrome (MetS) in the Chinese population. To examine the effect of the dietary pattern reflecting CRP with MetS, a cross-sectional study was based on the health checkup data from the Beijing MJ Health Screening Centers between 2008 and 2018. The CRP-related dietary pattern was derived from 17 food groups using reduced-rank regression. Participants were divided into five groups according to the quintiles of dietary pattern score. Multivariate logistic regression was then applied to estimate the odds ratios (OR) and 95% confidence intervals (CIs) for the quintiles of diet pattern score related to MetS and its four components. Of the 90,130 participants included in this study, 11,209 had MetS. A CRP-related dietary pattern was derived, characterized by a higher consumption of staple food, fresh meat, processed products, and sugar-sweetened beverages but a lower intake of honey and jam, fruits, and dairy products. Compared with participants in the lowest quintile (Q1), participants in the higher quintiles were associated with increased risks of MetS in a dose–response manner after adjustment for potential confounders (*p* for linear trend < 0.001), the ORs for Q2 to Q5 were 1.10 (95% CI: 1.02–1.19), 1.14 (95% CI: 1.05–1.22), 1.23 (95% CI: 1.15–1.33), and 1.49 (95% CI: 1.38–1.61), respectively. Moreover, the effects were stronger among individuals aged 50 years or older. A CRP-related dietary pattern was associated with the risk of MetS. It provides new insights that dietary intervention to achieve a lower inflammatory level could potentially prevent MetS.

## 1. Introduction

Metabolic syndrome (MetS) is a common chronic disease with a prevalence of approximately 33% in the United States [1], and 20% in China [2,3]. Meanwhile, MetS could increase the risk of cardiovascular disease, diabetes, and psychological disorders, resulting in a heavy health and economic burden all over the world [4,5,6].

Dietary patterns, which consist of a group of food components, are considered to have a significant influence on MetS [7,8]. Different dietary patterns are associated with varying risks of MetS [8], and different adherence to the same dietary pattern is also linked to different risks [7,8]. Hossein et al. indicated that participants who adhered to a mixed dietary pattern, including a combination of healthy (e.g., non-refined grains, vegetables) and unhealthy (e.g., mayonnaise, high-fat dairy products) foods, had a 2.68-fold higher risk of developing MetS compared with those who did not [8], but the underlying mechanisms were still unclear.

C-reactive protein (CRP) is one of the most important inflammatory markers which plays an essential role in the health effect of dietary patterns. Firstly, CRP levels in the body are influenced by dietary habits. A study from Italy found that a dietary pattern containing harmful food components was significantly associated with a higher level of CRP in the body [9], implying that the dietary pattern was associated with CRP. Secondly, elevated CRP is associated with MetS. Experimental studies have demonstrated that elevated CRP leads to multiple features of metabolic syndrome [10], suggesting CRP might be an intermediate involved in the development of MetS. Studies also showed that CRP contributed to abnormal endothelial function [11], resulting in MetS and cardiovascular disease outcomes [12]. A recent review showed that the dietary inflammatory index was associated with MetS [13], but evidence from Chinese adults is still unclear. Thus, it is essential to conduct a CRP-related dietary pattern study and assess the association of the dietary pattern with MetS in Chinese adults.

The reduced rank regression (RRR) could estimate pathway-dependent relationships based on the prior knowledge of underlying mediators between a dietary pattern and disease outcomes, and construct the dietary pattern with the maximum explanation of variation in independent variables concerning intermediates [14,15]. In the present study, we conducted this cross-sectional study using the RRR method to construct the dietary pattern reflecting the CRP level and estimate its association with the risk of MetS in Chinese adults.

## 2. Materials and Methods

### 2.1. Population

A cross-sectional study was designed based on the health checkup data from the Beijing MJ Health Screening Centers (MJHSC, Beijing, China) between 2008 and 2018. Participants were asked to finish a comprehensive questionnaire at each visit, including sociodemographic characteristics, lifestyle, dietary habits, and medical conditions, followed by physical examination and biochemistry tests.

In the present study, we included 97,106 adults with complete information at the first visit, considering the possible reverse effect of the health check on dietary habit and other behavioral factors in the subsequent visits. We further excluded individuals who reported a history of cancer (*n* = 587) or cardiovascular diseases (*n* = 2871), or implausible daily energy intake (women <500 or >3500 kcal/day; men <800 or >4200 kcal/ day) (*n* = 3518), leaving 90,130 participants for the main analysis. In RRR analysis, participants who had prevalent MetS (*n* = 11,209), with missing value on CRP (*n* = 58,320) or had acute inflammation (CRP > 10 μg/mL) (*n* = 1615) were further excluded, leaving 18,986 participants (Appendix A). The MJHSC data were held centrally by the Data Center. All the personal information in the database was removed prior to conducting the research. The study has been approved by the Institutional Review Board of Peking University Health Science Center (No. IRB00001052-19077).

### 2.2. Dietary Assessment

Dietary habits were assessed using a semi-quantitative food frequency questionnaire (FFQ), which collected the frequency and amount of food consumption over the past month. The details of FFQ in MJHSC have been described previously [16]. Briefly, it includes 17 dietary items, such as staple food, milk (including goat’s milk, yogurt, milk powder), dairy products (cheese), eggs, fresh meat (beef, mutton, pork, chicken, duck), fishery products, viscera, soy products, fruits, light green vegetables, dark green vegetables, root vegetables, honey and jam, sugar-sweetened beverages, fried foods, processed products, and salt seasoning. For each food item, the portion size was defined with examples. The frequency had five options (never or fewer than 1 portion per week, 1 to 3 portions per week, 4 to 6 portions per week, 1 portion a day, at least 2 portions each day) for each food item.

### 2.3. Covariate Assessment

Sociodemographic status (age, sex, marital status, education, occupation, and annual income), lifestyle habits (physical leisure activity, alcohol drinking, and smoking status), and medical conditions (previously diagnosis of hypertension or diabetes, medication usage, family history of hypertension or diabetes) were also gathered using paper-based questionnaires. The physical leisure activity was calculated from intensity (mild, moderate, severe, and vigorous), frequency (times per day), and duration (hours per time) of the type of physical activity and summing the metabolic equivalent hours per day (MET-h/day). The values for mild, moderate, severe, and vigorous physical activity were 1.5, 4.5, 7.5, and 10.5 MET, respectively. Alcohol drinking and smoking were categorized as “never or occasional”, “former regular”, and “current regular”.

In the physical examination, staff measured waist circumference (WC), systolic blood pressure (SBP), and diastolic blood pressure (DBP) with standard procedures. WC was measured at the end of a normal expiration at the level of noticeable waist narrowing between the lowest aspect of the rib cage and the highest point of the iliac crest. When narrowing could not be determined, the circumference was measured at the level of the umbilicus. SBP and DBP were measured using an automatic mercury sphygmomanometer (Citizen CH-5000, Tokyo, Japan) twice in the right arm after 10 min rest, and the mean values were taken.

Blood samples were collected by trained nurses. Fasting plasma glucose (FPG), total triglyceride (TG), high-density lipoprotein cholesterol (HDL-C), and C-reactive protein (CRP) were tested by an automatic biochemical analyzer (Olympus Au 1000, Tokyo, Japan).

### 2.4. Definition of MetS

According to the Joint Committee for Developing Chinese Guidelines on Prevention and Treatment of Dyslipidemia in Adults (JCDCG) [17], the MetS was defined as having three or more of the five following conditions: (i) abdominal obesity: WC ≥ 90 cm in male or ≥ 85 cm in female; (ii) hyperglycemia: FPG ≥ 6.1 mmol/L and/or 2-h plasma glucose (2 h-PG) ≥ 7.8 mmol/L and/or diagnosed with diabetes and/or treated for diabetes; (iii) high blood pressure: SBP ≥ 130 mmHg or DBP ≥ 85 mmHg and/or diagnosed with hypertension and/or treated for hypertension; (iv) TG ≥ 1.7 mmol/L; (v) HDL-C < 1.04 mmol/L. Participants who meet criteria vi and v (abnormal TG or HDL-C) were defined as having hyperlipidemia.

### 2.5. Statistical Analysis

Food frequency was assigned to the middle value of each option and z-standardized. We used 17 dietary items as predictive variables and the log-transformed CRP as the response variable in the RRR analysis and derived a dietary pattern that could explain the maximum variation of log-transformed CRP. We calculated the dietary pattern score for all participants by summing the product of z-standardized intakes and weights of 17 dietary items. The dietary pattern was categorized according to quintiles, with the group of lowest quintiles (Q1) representing those who least conformed to the dietary pattern, and used as the reference group in the subsequent analysis.

Linear regressions and logistic regressions were used to describe the means and percentages of sociodemographic, behavioral characteristics, and family history of hypertension and diabetes according to quintiles, adjusted for age and sex as appropriate. Multivariate logistic regression was applied to estimate the odds ratios (OR) and 95% confidence interval (95% CI) for the quintiles of diet pattern score with MetS and its four components (abdominal obesity, hyperglycemia, high blood pressure, hyperlipidemia), taking the lowest quintile (Q1) as the reference. We adjusted for age, sex, and daily energy intake in Model 1. In Model 2, we additionally adjusted for marital status, education, annual income, family history of hypertension, diabetes, and behavioral characteristics (physical activity, smoking status, and alcohol drinking). Linear trend tests were performed by assigning the midpoint values of the diet pattern and treating the variable as continuous in a separate regression model.

In subgroup analysis, we estimated the association between the dietary pattern and MetS stratified by age (<50, ≥50), sex (male, female), alcohol drinking (never, ever), smoking status (never, ever), and physical leisure activity (below median, above median). We evaluated the interaction between the dietary pattern and these covariates by likelihood ratio test comparing the models with and without the cross-product terms. Two sensitivity analyses were conducted separately: (1) excluding those with a missing value on physical leisure activity (*n* = 21,607); or (2) additionally adjusting for occupation.

All statistical analyses were performed using the SAS software (version 9.4, SAS Institute, Cary, NC, USA). We defined *p* < 0.05 as statistically significant.

## 3. Results

### 3.1. CRP-Related Dietary Pattern and Its Characteristics

Among the participants eligible for the RRR analysis (*n* = 18,986), the derived dietary pattern could explain 1.02% of the total variation in serum CRP level. As shown in Figure 1, the dietary pattern was characterized by higher consumption of staple food, fresh meat, processed products, and sugar-sweetened beverages but a lower intake of honey and jam, fruits, and dairy products.

As shown in Table 1, participants who followed the dietary pattern, i.e., with a higher dietary pattern score, were more likely to be younger, male, alcohol drinkers, cigarette smokers, to have higher annual income, more daily energy intake, and were less physically active, more likely to report family histories of hypertension or diabetes (*p* < 0.05). In addition, the CRP concentration was positively related to the dietary pattern scores quintile (shown in Appendix A).

### 3.2. Association of CRP-Related Dietary Pattern with MetS

Of the 90,130 participants, 11,209 (12.4%) individuals were diagnosed with MetS, including 19,164 (21.3%) cases of abdominal obesity, 10,221 (11.3%) cases of hyperglycemia, 15,495 (17.2%) cases of high blood pressure, and 40,852 (45.3%) cases of hyperlipidemia.

After adjustment for potential confounders, an inverse dose–response relationship was estimated between the CRP-related dietary pattern and MetS (*P*_trend_ < 0.001, shown in Table 2). Compared with the lowest quintile (Q1), participants in the highest quintile (Q5) had the highest risk of MetS (OR: 1.49; 95% CI: 1.38,1.61). A similar dose–response relationship was observed for 3 components of MetS (i.e., abdominal obesity, hyperglycemia, high blood pressure; all *P*_trend_ < 0.001). The association remained unchanged after excluding individuals with a missing value on physical leisure activity or adjusted for occupation in the sensitivity analyses (Appendix A).

In the subgroup analysis by age, sex, alcohol consumption, cigarette smoking, and physical leisure activity level (Table 3), no significant interactions with these factors were observed except age group (*P* for interaction < 0.001). Among participants aged ≥50 years, stronger associations of the CRP-related dietary pattern with MetS were estimated than those in the younger adults.

## 4. Discussion

This cross-sectional study derived a CRP-related dietary pattern characterized by higher consumption of staple food, fresh meat, processed products, and sugar-sweetened beverages but a lower intake of honey and jam, fruits, and dairy products. The dietary pattern was associated with an increased risk of MetS in a dose–response manner after adjusting potential confounders, especially in older adults over 50 years. Our study provided further evidence that the dietary pattern would affect MetS through CRP level and highlights the potential public health implications of MetS prevention through anti-inflammatory diets. As a modifiable factor in daily life, dietary inventions based on our results could prevent MetS development among high-risk individuals, i.e., reducing the CRP-related dietary pattern by increasing the intake of healthy food (honey and jam, fruits, and dairy products), and decreasing consumption of unhealthy food (staple food, fresh meat, processed products, and sugar-sweetened beverages).

### 4.1. Comparison with Previous Studies

In the present study, we selected CRP as an intermediate pathway in the association between diet and MetS. Previous studies had shown that dietary pattern could influence serum level of CRP [9], and high serum CRP level is known widely as an established risk factor in MetS [10,13]. The SU.VI.MAX cohort study including 3726 participants found that the participants with the highest Dietary Inflammatory Index (DII) score had a 1.39-time higher risk of developing the MetS than those with the lowest [18]. A study among Korean adults established a high-sensitivity C-reactive protein (hs-CRP)-related dietary pattern score using Spearman correlation and multiple regression, and also found that the highest score was positively associated with a 2.618-fold higher prevalence of MetS than the lowest [19]. In a cross-sectional study, the dietary patterns characterized by low intake of olive oil, vegetables, legumes, soups, fruits, fish, and high intake of red meat, animal fats, alcohol, which were generated using principal factor analysis, were associated with both CRP and cardiovascular disease risk profile in the health [9]. Additionally, the participants with the highest quartile of hs-CRP were reported to have a 1.83-fold higher risk than those with the lowest in a previous cohort study [20], which supported the findings of the current study. Therefore, it was of public health significance to identify the dietary patterns that could affect CRP levels and therefore lower the risk of MetS.

The dietary pattern we derived in this study could reflect the CRP level effectively. In the present study, the dietary pattern explained 1.02% of the total variation in serum CRP level, similar to the previously reported effect size [21]. The present study indicates that fruits and light/dark green vegetables are associated with lower CRP concentrations, consistent with previous research. A meta-analysis involving 26 observational studies showed that a vegetarian-based dietary pattern was significantly associated with a lower CRP [22]. Moreover, the low CRP-related dietary pattern found in this study shared many commonalities of food components with the Mediterranean diet. As a well-known dietary pattern, the Mediterranean diet, with its regular intake of vegetables, fruits, and fish, is considered to be a dietary pattern that reduces serum CRP concentrations. Arouca et al. found that participants with a higher adherence to the Mediterranean diet had a lower CRP concentration [23]. Honey and jam, which were reported to be protective food components for MetS in previous studies [24,25], had the largest negative loading in this study. Honey, a natural compound with a range of biological components, such as flavonoids, phenolic acids, antioxidant enzymes, ascorbic acid, and carotenoid [26], has been shown to have antioxidant and anti-inflammation effects [27], thus further reducing the risk of MetS.

In the subgroups analysis, we found that the CRP-related dietary pattern had a stronger protective effect on MetS in the participants ≥50 years old, indicating the elderly could be a targeted population to adapt the CRP-related dietary pattern to prevent MetS.

Previous epidemiological studies reported that one-fourth of US adults over 65 years old had a higher risk of MetS [28]. Moreover, they had a higher CRP level than the younger adults. The US Health and Retirement Study reported that an older age was significantly associated with a higher level of CRP [29]. In particular, negative life events, usually more common in the elderly, could raise the CRP level, such as unemployment [30]. Therefore, our results suggest new dietary interventions would be effective in preventing MetS, especially for the older population.

### 4.2. Strengths and Limitations

Our results benefited from the large sample size and various controlled confounders. Unlike data-driven methods (e.g., factor analysis, clustering analysis), the RRR approach enabled us to use prior knowledge of CRP on MetS to derive dietary patterns. Few studies utilize pathway information to construct dietary patterns in the Chinese population. However, several limitations should be mentioned. First, the nature of the cross-sectional study design limited the ability to infer causality, and prospective studies are still needed. Second, the health checkup took place in downtown Beijing, and the study population had higher income and education levels, so the extrapolation of our results should be cautious. Third, CRP is one of the critical inflammatory indices, but we did not utilize multiple biomarkers as response variables due to their availability in health examination data. Therefore, we may not be able to depict the overall inflammatory status of each participant. Fourth, the residual confounding could not be excluded even though we adjusted a series of potential confounders. Some diseases that could affect the level of CRP, such as infectious diseases, rheumatic diseases, connective tissue diseases, etc., were not considered in the present study. Further research adjusting for more comprehensive covariates is still needed.

## 5. Conclusions

A CRP-related dietary pattern, characterized by higher consumption of staple food, fresh meat, processed products, sugar-sweetened beverages, and a lower intake of honey and jam, fruits, and dairy products, was associated with a higher risk of MetS in the Chinese population. It provided evidence for dietary guidelines for MetS prevention by following anti-inflammatory diet habits, especially among elderly adults.

## Figures and Tables

**Figure 1 nutrients-14-02566-f001:**
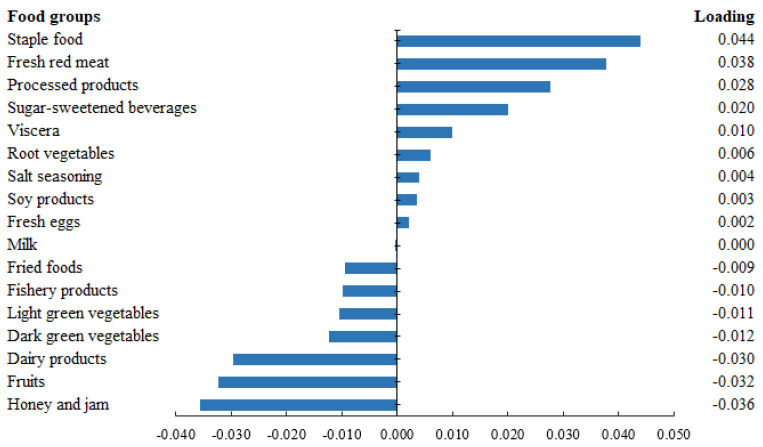
Factor loadings of dietary patterns derived from reduced rank regression (RRR) in the Chinese population of MJ Health Database (*n* = 18,986). The length of the bar shows the loading of specific food components on C-reactive protein (CRP)-related dietary pattern. The bar indicates a positive load to the right and a negative load to the left.

**Table 1 nutrients-14-02566-t001:** Characteristics of participants according to quintiles of dietary pattern scores in MJHSC Database (*n* = 90,130).

Characteristics	Dietary Pattern Scores				
	Q1	Q2	Q3	Q4	Q5
Age (years, $\bar{x}$ ± SD)	39.8 ± 11.5	40.4 ± 11.5	39.6 ± 11.2	39.2 ± 11.0	38.1 ± 10.3
Male (%)	30.3	41.4	51.2	63.9	75.8
Married (%)	80.3	82.9	82.9	83.1	81.9
College and above (%)	92.4	90.6	90.5	91.2	91.2
Annual income (%)					
<100,000	40.0	41.2	40.4	38.5	37.6
100,000–199,999	32.0	32.3	32.8	33.0	32.7
≥200,000	28.0	26.5	26.9	28.6	29.7
Physical leisure activity (MET-h/day, $\bar{x}$ ± SD) *	4.1 ± 0.5	3.5 ± 0.5	3.3 ± 0.5	3.1 ± 0.5	2.8 ± 0.5
Alcohol consumption (%)					
Never	78.4	74.7	70.6	64.6	57.1
Former drinker	2.6	2.9	2.8	3.0	3.0
Current drinker	20.0	22.4	26.4	32.4	39.9
Smoking (%)					
Never	89.2	84.8	79.9	73.3	63.5
Former smoker	3.1	3.4	3.9	4.9	5.0
Current smoker	7.7	11.6	16.1	21.8	31.5
Daily energy intake (kcal/day, $\bar{x}$ ± SD)	1085.8 ± 264.9	1066.7 ± 235.1	1114.9 ± 242.2	1186.7 ± 246.8	1350.0 ± 285.2
Family History of HTN (%)	38.2	38.9	39.4	39.99	41.3
Family History of DM (%)	22.4	22.1	21.8	22.6	23.6

Q indicates quintile; HTN indicates hypertension; DM indicates diabetes. Values are means or percentages of participants adjusted for age and sex as appropriate. * Among 68,523 participants with physical activity information.

**Table 2 nutrients-14-02566-t002:** Associations between dietary pattern and metabolic syndrome (MetS), as well as its four components. (*n* = 90,130).

	OR (95% CI)					*P* _trend_
	Q1	Q2	Q3	Q4	Q5	
MetS						
Cases (*n*)	1374	1836	2167	2558	3274	
Model 1	1.00	1.15 (1.06, 1.24)	1.23 (1.14, 1.32)	1.38 (1.28, 1.48)	1.77 (1.64, 1.90)	<0.001
Model 2	1.00	1.10 (1.02, 1.19)	1.14 (1.05, 1.22)	1.23 (1.15, 1.33)	1.49 (1.38, 1.61)	<0.001
Abdominal obesity						
Cases (*n*)	2542	3249	3668	4352	5353	
Model 1	1.00	1.15 (1.08, 1.22)	1.21 (1.14, 1.28)	1.33 (1.25, 1.41)	1.64 (1.55, 1.74)	<0.001
Model 2	1.00	1.11 (1.05, 1.18)	1.14 (1.07, 1.21)	1.22 (1.15, 1.29)	1.45 (1.36, 1.54)	<0.001
Hyperglycemia						
Cases (*n*)	1469	1871	2050	2314	2517	
Model 1	1.00	1.18 (1.10, 1.28)	1.32 (1.22, 1.42)	1.46 (1.35, 1.58)	1.72 (1.59, 1.86)	<0.001
Model 2	1.00	1.16 (1.07, 1.25)	1.26 (1.17, 1.36)	1.35 (1.25, 1.46)	1.52 (1.40, 1.65)	<0.001
High blood pressure						
Cases (*n*)	2433	2933	3104	3374	3651	
Model 1	1.00	1.11 (1.04, 1.19)	1.16 (1.09, 1.24)	1.17 (1.10, 1.24)	1.28 (1.20, 1.37)	<0.001
Model 2	1.00	1.09 (1.02, 1.16)	1.11 (1.04, 1.19)	1.11 (1.04, 1.18)	1.19 (1.11, 1.28)	<0.001
Hyperlipidemia						
Cases (*n*)	7243	7548	7993	8553	9515	
Model 1	1.00	0.98 (0.93, 1.02)	0.98 (0.94, 1.02)	1.00 (0.96, 1.05)	1.09 (1.05, 1.16)	<0.001
Model 2	1.00	0.96 (0.92, 1.00)	0.95 (0.91, 0.99)	0.95 (0.91, 1.00)	1.00 (0.96, 1.06)	0.74

OR, odds ratio; CI, confidence interval; Q indicates quintile; MetS indicates metabolic syndrome. Reference group: the lowest quintile (Q1) of dietary pattern scores. ORs and its 95% CIs were obtained using a logistic regression model. Model 1 was adjusted for age, sex, and daily intake of energy; Model 2 was adjusted for education, annual income, marital status, family history of hypertension, diabetes, physical leisure activity, alcohol use, and smoking status.

**Table 3 nutrients-14-02566-t003:** Subgroup analysis of the associations between dietary pattern and metabolic syndrome (MetS).

	OR (95% CI)					*P* _i_ _nteract_
	Q1	Q2	Q3	Q4	Q5	
Age (years)						<0.001
<50	1.00	1.08 (0.98, 1.20)	1.07 (0.97, 1.18)	1.19 (1.08, 1.31)	1.44 (1.31, 1.58)	
≥50	1.00	1.11 (0.99, 1.25)	1.27 (1.13, 1.43)	1.32 (1.17, 1.50)	1.44 (1.26, 1.65)	
Sex						0.38
Male	1.00	1.03 (0.94, 1.13)	1.06 (0.97, 1.16)	1.13 (1.04, 1.23)	1.34 (1.23, 1.46)	
Female	1.00	1.17 (1.01, 1.34)	1.23 (1.07, 1.43)	1.42 (1.21, 1.66)	1.79 (1.49, 2.15)	
Alcohol						0.09
Never	1.00	1.12 (1.01, 1.23)	1.18 (1.07, 1.31)	1.22 (1.11, 1.35)	1.51 (1.36, 1.67)	
Ever	1.00	1.06 (0.93, 1.20)	1.07 (0.94, 1.21)	1.22 (1.08, 1.37)	1.45 (1.29, 1.62)	
Smoking						0.46
Never	1.00	1.10 (0.99, 1.21)	1.19 (1.08, 1.31)	1.24 (1.13, 1.37)	1.59 (1.43, 1.77)	
Ever	1.00	1.09 (0.96, 1.24)	1.07 (0.95, 1.21)	1.21 (1.07, 1.36)	1.38 (1.23, 1.56)	
Physical leisure activity level *						0.28
Low	1.00	1.19 (1.06, 1.35)	1.21 (1.07, 1.37)	1.38 (1.23, 1.56)	1.64 (1.45, 1.85)	
High	1.00	1.03 (0.90, 1.17)	1.08 (0.96, 1.22)	1.11 (0.98, 1.26)	1.34 (1.19, 1.52)	

OR, odds ratio; CI, confidence interval; Q indicates quintile. Reference group: the lowest quintile of dietary pattern scores. ORs and its 95% CIs were obtained using logistic regression model. All results were adjusted for age, sex, daily intake of energy, education, annual income, marital status, family history of hypertension and diabetes, physical activity, alcohol use and smoking status. * Physical leisure activity was divided into two groups (Low and High) by the median of the metabolic equivalent hours per day (MET-h/day).

## Data Availability

The datasets analyzed during the current study are not publicly available due to the protection of privacy considering the ethics but are available from the corresponding author on reasonable request.

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
