# Peer review of "Association between Dietary Patterns Reflecting C-Reactive Protein and Metabolic Syndrome in the Chinese Population"

_nutrients, 2022, doi:10.3390/nu14132566_

Round 1
Reviewer 1 Report
Article
Association between dietary patterns based on reduced rank regression and metabolic syndrome in the Chinese population
- This study measures the associations between CRP levels with the risk of metabolic syndrome in Chinese adults.
- The article is well written and designed properly.
- The study answered the research question and provides a valuable data regarding.
Important notes
Title: the current title is not suitable and did not match with the flow of the study.
I am suggesting change of title of the study because this title did not reflect the findings of the study and the work which has been done.
Abstract:
- Authors have to write down the objective of the study in the abstract and rewrite it structurally.
- Introduction:
- Well written
- Methodology:
- Well written
- Results:
- Well presented
- Discussion:
- Authors should add some studies explored the association of CRP with the risk of metabolic syndrome to compare it with the finding of the current study.
Conclusion
- Conclusion should be rewritten properly to reflect the aim and the findings of the study.
References
- References should be written according to the journal guidelines as most of them are not according to the journal instructions. For example, date should be bold.
Author Response
Dear Ms. Fifteen Hu and reviewers,
Thank you so much for your precious comments and advice concerning our manuscript entitled “Association between dietary patterns based on reduced rank regression and metabolic syndrome in the Chinese population”. The comments are very valuable and helpful. We have read the comments carefully and uploaded files of the revised manuscript. Revisions in the text are highlighted in ‘track changes’ and the responses to the comments are as follows.
Reviewer 1:
Q1: Title: the current title is not suitable and did not match with the flow of the study. I am suggesting change of title of the study because this title did not reflect the findings of the study and the work which has been done.
A1: We are grateful for your suggestion. We changed our title to “Association between dietary patterns reflecting C-reactive protein and metabolic syndrome in the Chinese population”, in order to highlight the importance of the CRP-related dietary patterns in our findings. (lines 2-4)
Q2: Abstract: Authors have to write down the objective of the study in the abstract and rewrite it structurally.
A2: Thank you for your precious advice. We noticed that the abstract should consist of four parts (background, methods, results, and conclusion) as the instructions for authors. The objective was added in the background part of the abstract. (lines 16-17)
Q3: Discussion: Authors should add some studies explored the association of CRP with the risk of metabolic syndrome to compare it with the finding of the current study.
A3: We are grateful for your valuable suggestion. We further discussed the association of CRP with the risk of metabolic syndrome in previous studies, and the association reported before also supported our results. More details can be found in lines 245-248.
Q4: Conclusion: Conclusion should be rewritten properly to reflect the aim and the findings of the study.
A4: We agree with the comments and adjusted the conclusion part. More details can be found in lines 294-299.
Q5: References should be written according to the journal guidelines as most of them are not according to the journal instructions. For example, date should be bold.
A5: We apologize for the reference problem in the manuscript. Date of the reference has been revised to be bold, and Endnote was used to manage the references.

Reviewer 2 Report
This is a good study on the relationship between dietary patterns reflecting CRP and metabolic syndrome.
Title: It is difficult to say that the reduced rank analysis, which is mostly performed in regression analysis, has special significance. Therefore, it seems better to omit reduced rank regression from the title and instead describe it as diatary patterns reflecting CRP rather than dietary patterns.
Line 40-41: add [ ] to the reference number(s).
Line 75-76: Not all cardiovascular diseases increase CRP. It only increases in cases such as acute myocardial infarction. A specific description of cardiovascular disease is required. In addition to cancer and cardiovascular diseases, CRP increases in infectious diseases, rheumatic diseases, and connective tissue diseases. Therefore, it is necessary to evaluate such diseases affecting CRP and describe how to correct them. Any disease that cannot be evaluated should be described as a limitation.
Please briefly describe your statistical method for the reader at the bottom of Table 2 and 3. And please add the described statistical method to 2.5 Statistical analysis.
Table 1: Table 1 is simply a description. Please do a statistical analysis of whether there is a significant difference between the Q1 - Q5 groups.
Table 3: Please explain the definitions of low and high for physical activity below in Table 3.
Author Response
Dear Ms. Fifteen Hu and reviewers,
Thank you so much for your precious comments and advice concerning our manuscript entitled “Association between dietary patterns based on reduced rank regression and metabolic syndrome in the Chinese population”. The comments are very valuable and helpful. We have read the comments carefully and uploaded files of the revised manuscript. Revisions in the text are highlighted in ‘track changes’ and the responses to the comments are as follows.
Reviewer 2:
Q1: Title: It is difficult to say that the reduced rank analysis, which is mostly performed in regression analysis, has special significance. Therefore, it seems better to omit reduced rank regression from the title and instead describe it as diatary patterns reflecting CRP rather than dietary patterns.
A1: We are very grateful for your precious suggestion. We changed our title to “Association between dietary patterns reflecting C-reactive protein and metabolic syndrome in the Chinese population”, in order to highlight the importance of the CRP-related dietary patterns in our findings. (lines 2-4)
Q2: Line 40-41: add [ ] to the reference number(s).
A2: We apologize for the format problem in the manuscript. We have corrected the problem in lines 41-43, and checked references again.
Q3: Line 75-76: Not all cardiovascular diseases increase CRP. It only increases in cases such as acute myocardial infarction. A specific description of cardiovascular disease is required. In addition to cancer and cardiovascular diseases, CRP increases in infectious diseases, rheumatic diseases, and connective tissue diseases. Therefore, it is necessary to evaluate such diseases affecting CRP and describe how to correct them. Any disease that cannot be evaluated should be described as a limitation.
A3: Thank you for your valuable and professional comments. We agree that not all cardiovascular diseases (CVD) increase CRP. Actually, we excluded all participants with history of all CVD, and it was due to the consideration that the lifestyles and habits, including dietary patterns, might change after the onset of CVD. That is to say, participants who suffered from CVD might have a relatively healthy dietary pattern but were at a higher risk of MetS. It brings potential confounding and would make our research underestimate the association of the CRP-related dietary patterns with MetS. Therefore, we excluded all participants with history of all CVD to reduce the potential bias. Moreover, participants who had acute inflammatory (CRP > 10μg/mL) were also excluded in our study for the similar consideration.
Infectious diseases, rheumatic diseases, and connective tissue diseases affect the level of CRP indeed. We have added it as a limitation of our research in line 290-292, and further study adjusting for those covariates is still needed.
Thanks again for your comments.
Q4: Please briefly describe your statistical method for the reader at the bottom of Table 2 and 3. And please add the described statistical method to 2.5 Statistical analysis.
A4: Thank you for your careful reading of our manuscript. We used Multivariate logistic regression model to estimate the ORs and 95%CIs in Table 2 and 3. We have added the statistical method below the tables (Lines 199-200 and Lines 213-214), and the statistical method was described in lines 141-150.
Q5: Table 1: Table 1 is simply a description. Please do a statistical analysis of whether there is a significant difference between the Q1 - Q5 groups.
A5: Thank you so much for your suggestion. However, tests of the difference between quintiles were less important due to the large sample size (n=97,106). All p values were less than 0.001 for tests of all characteristic variables, so we only presented the means and percentages of the variables according to the quintiles of the dietary pattern. If the reviewer insisted add p-values, we will add a statement below Table 1.
Q6: Table 3: Please explain the definitions of low and high for physical activity below in Table 3.
A6: Thank you for your suggestion. We have added the definitions of low and high for physical activity below in Table 3. (lines 217-218)
Moreover, the self-citations of Nutrients have been reduced as suggested by the editor. Many thanks for your efforts and careful review. Please do not hesitate to contact me if you have any further questions.
